# Teaching Strategies for Developing Clinical Reasoning Skills in Nursing Students: A Systematic Review of Randomised Controlled Trials

**DOI:** 10.3390/healthcare12010090

**Published:** 2023-12-30

**Authors:** Ana Pérez-Perdomo, Adelaida Zabalegui

**Affiliations:** Hospital Clinic of Barcelona, 08036 Barcelona, Spain; anperez@clinic.cat

**Keywords:** nursing student, clinical reasoning, clinical decision making, thinking skills, randomised controlled trials

## Abstract

Background: Clinical reasoning (CR) is a holistic and recursive cognitive process. It allows nursing students to accurately perceive patients’ situations and choose the best course of action among the available alternatives. This study aimed to identify the randomised controlled trials studies in the literature that concern clinical reasoning in the context of nursing students. Methods: A comprehensive search of PubMed, Scopus, Embase, and the Cochrane Controlled Register of Trials (CENTRAL) was performed to identify relevant studies published up to October 2023. The following inclusion criteria were examined: (a) clinical reasoning, clinical judgment, and critical thinking in nursing students as a primary study aim; (b) articles published for the last eleven years; (c) research conducted between January 2012 and September 2023; (d) articles published only in English and Spanish; and (e) Randomised Clinical Trials. The Critical Appraisal Skills Programme tool was utilised to appraise all included studies. Results: Fifteen papers were analysed. Based on the teaching strategies used in the articles, two groups have been identified: simulation methods and learning programs. The studies focus on comparing different teaching methodologies. Conclusions: This systematic review has detected different approaches to help nursing students improve their reasoning and decision-making skills. The use of mobile apps, digital simulations, and learning games has a positive impact on the clinical reasoning abilities of nursing students and their motivation. Incorporating new technologies into problem-solving-based learning and decision-making can also enhance nursing students’ reasoning skills. Nursing schools should evaluate their current methods and consider integrating or modifying new technologies and methodologies that can help enhance students’ learning and improve their clinical reasoning and cognitive skills.

## 1. Introduction

Clinical reasoning (CR) is a holistic cognitive process. It allows nursing students to accurately perceive patients’ situations and choose the best course of action among the available alternatives. This process is consistent, dynamic, and flexible, and it helps nursing students gain awareness and put their learning into perspective [1]. CR is an essential competence for nurses’ professional practice. It is considered crucial that its development begin during basic training [2]. Analysing clinical data, determining priorities, developing plans, and interpreting results are primary skills in clinical reasoning during clinical nursing practise [3]. To develop these skills, nursing students must participate in caring for patients and working in teams during clinical experiences. Among clinical reasoning skills, we can identify communication skills as necessary for connecting with patients, conducting health interviews, engaging in shared decision-making, eliciting patients’ concerns and expectations, discussing clinical cases with colleagues and supervisors, and explaining one’s reasoning to others [4].

Educating students in nursing practise to ensure high-quality learning and safe clinical practise is a constant challenge [5]. Facilitating the development of reasoning is challenging for educators due to its complexity and multifaceted nature [6], but it is necessary because clinical reasoning must be embedded throughout the nursing curriculum [7]. Such being the case, the development of clinical reasoning is encouraged, aiming to promote better performance in indispensable skills, decision-making, quality, and safety when assisting patients [8].

Nursing education is targeted at recognising clinical signs and symptoms, accurately assessing the patient, appropriately intervening, and evaluating the effectiveness of interventions. All these clinical processes require clinical reasoning, and it takes time to develop [9]. This is a significant goal of nursing education [10] in contemporary teaching and learning approaches [6].

Strategies to mitigate errors, promote knowledge acquisition, and develop clinical reasoning should be adopted in the training of health professionals. According to the literature, different methods and teaching strategies can be applied during nursing training, as well as traditional teaching through lectures. However, the literature explains that this type of methodology cannot enhance students’ clinical reasoning alone. Therefore, nursing educators are tasked with looking for other methodologies that improve students’ clinical reasoning [11], such as clinical simulation. Clinical simulation offers a secure and controlled setting to encounter and contemplate clinical scenarios, establish relationships, gather information, and exercise autonomy in decision-making and problem-solving [12]. Different teaching strategies have been developed in clinical simulation, like games or case studies. Research indicates a positive correlation between the use of simulation to improve learning outcomes and how it positively influences the development of students’ clinical reasoning skills [13].

The students of the 21st century utilise information and communication technologies. With their technological skills, organisations can enhance their productivity and achieve their goals more efficiently. Serious games are simulations that use technology to provide nursing students with a safe and realistic environment to practise clinical reasoning and decision-making skills [14] and can foster the development of clinical reasoning through an engaging and motivating experience [15].

New graduate nurses must possess the reasoning skills required to handle complex patient situations. Aware that there are different teaching methodologies, with this systematic review we intend to discover which RCTs published focus on CR in nursing students, which interventions have been developed, and their effectiveness, both at the level of knowledge and in increasing clinical reasoning skills. By identifying the different techniques used during the interventions with nursing students in recent years and their effectiveness, it will help universities decide which type of methodology to implement to improve the reasoning skills of nursing students and, therefore, obtain better healthcare results.

This study aims to identify and analyse randomised controlled trials concerning clinical reasoning in nursing students. The following questions guide this literature review:

Which randomised controlled trials have been conducted in the last eleven years regarding nursing students’ clinical reasoning? What are the purposes of the identified RCTs? Which teaching methodologies or strategies were used in the RCTs studies? What were the outcomes of the teaching strategies used in the RCTs?

## 2. Materials and Methods

This review follows the PRISMA 2020 model statement for systematic reviews. That comprises three documents: the 27-item checklist, the PRISMA 2020 abstract checklist, and the revised flow diagram [16].

### 2.1. Search Strategy

A systematic literature review was conducted on PubMed, Scopus, Embase, and the Cochrane Controlled Register of Trials (CENTRAL) up to 15th October 2023.

The PICOS methodology guided the bibliographic search [17]: “P” being the population (nursing students), “I” the intervention (clinical reasoning), “C” comparison (traditional teaching), “O” outcome (dimension, context, and attributes of clinical reasoning in the students’ competences and the results of the teaching method on nursing students), and “S” study type (RCTs).

The search strategy used in each database was the following: (“nursing students” OR “nursing students” OR “pupil nurses” OR “undergraduate nursing”) AND (“clinical reasoning” OR “critical thinking” OR “clinical judgment”). The filters applied were full text, randomised controlled trial, English, Spanish, and from 1 January 2012 to 15 October 2023. The search strategy was performed using the same process for each database. APP performed the search, and AZ supervised the process.

During the search, the terms clinical reasoning, critical thinking, and clinical judgement were used interchangeably since clinical judgement is part of clinical reasoning and is defined by the decision to act. It is influenced by an individual’s previous experiences and clinical reasoning skills [18]. Critical thinking and clinical judgement involve reflective and logical thinking skills and play a vital role in the decision-making and problem-solving processes [19].

The first search was conducted between March and September 2022, and an additional search was conducted during October 2023, adding the new articles published between September 2022 and September 2023, following the same strategy. The search strategy was developed using words from article titles, abstracts, and index terms. Parallel to this process, the PRISMA protocol was used to systematise the collection of all the information presented in each selected article. This systematic review protocol was registered in the international register PROSPERO: CRD42022372240.

### 2.2. Eligibility Criteria and Study Selection

The following inclusion criteria were examined: (a) clinical reasoning, clinical judgment, and critical thinking in nursing students as a primary aim; (b) articles published in the last eleven years; (c) research conducted between January 2012 and September 2023; (d) articles published only in English and Spanish; and (e) RCTs. On the other hand, the exclusion criteria were studies conducted with students from other disciplines other than nursing, not random studies or review articles.

### 2.3. Data Collection and Extraction

After this study selection, the following information was extracted from each article: bibliographic information, study aims, teaching methodology, sample size and characteristics, time of intervention, and conclusions.

### 2.4. Risk of Bias

The two reviewers, APP and AZ, worked independently to minimise bias and mistakes. The titles and abstracts of all papers were screened for inclusion. All potential articles underwent a two-stage screening process based on the inclusion criteria. All citations were screened based on title, abstract, and text. Reviewers discussed the results to resolve minor discrepancies. All uncertain citations were included for full-text review. The full text of each included citation was obtained. Each study was read thoroughly and assessed for inclusion following the inclusion and exclusion criteria explained in the methodology. The CASP tool was utilised to appraise all included studies. The CASP Randomized Controlled Trial Standard Checklist is an 11-question checklist [20], and the components assessed included the appropriateness of the objective and aims, methodology, study design, sampling method, data collection, reflexivity of the researchers, ethical considerations, data analysis, rigour of findings, and significance of this research. These items of the studies were then rated (“Yes” = with three points; “Cannot tell” = with two points; “No” = with one point). The possible rates for every article were between 0 and 39 points.

### 2.5. Ethical Considerations

Since this study was a comprehensive, systematic review of the existing published literature, there was no need for us to seek ethical approval.

## 3. Results

### 3.1. Search Results

The initial search identified 158 articles using the above-mentioned strategy (SCOPUS^®^ n = 72, PUBMED^®^ n = 56, CENTRAL^®^ n = 23, and EMBASE^®^ n= 7), and the results are presented in Figure 1. After retrieving the articles and excluding 111, 47 were selected for a full reading. Finally, 17 articles were selected. To comply with the methodology, the independent reviewers analysed all the selected articles one more time after the additional search, and they agreed to eliminate two of them because this study sample included nursing students as well as professional nurses. Therefore, to have a clear outcome focused on nursing students, two articles were removed, and the very final sample size was fifteen articles, following the established selection criteria (Figure 1). The reasons for excluding studies from the systematic review were: nurses as targets; other design types of studies different from RCTs; focusing on other health professionals such as medical students; review studies; and being published in full text in other languages other than Spanish or English.

### 3.2. Risk of Bias in CASP Results

All studies included in the review were screened with the CASP tool. Each study was scored out of a maximum of 39 points, showing the high quality of the randomised control trial methodology. The studies included had an average score of 33.1, ranging from 30 to 36 points. In addition, this quantitative rate of the items based on CASP, there were 13 studies that missed an item in relation to assessing/analysing outcome/s ‘blinded or not’ or not, and 11 studies that missed the item whether the benefits of the experimental intervention outweigh the harms and costs.

### 3.3. Data Extraction

Once the articles had undergone a full reading and the inclusion criteria were applied, data extraction was performed with a data extraction table (Appendix A). Their contents were summarised into six different cells: (1) CASP total points result, (2) purpose of this study, (3) teaching strategy, (4) time of intervention, (5) sample size, and (6) author and year of publication. After the review by the article’s readers, fifteen RCTs were selected. Of the fifteen, the continent with the highest number of studies was Asia, with 53.33% of the studies (n = 8) (Korea n = 4, Taiwan n = 2, and China n = 2), followed by Europe with 26.66% (n = 4) (Turkey n = 2, Paris n = 1, and Norway n = 1), and lastly South America with 20% (n = 3), all of them from Brazil.

### 3.4. Teaching Strategies

Different teaching strategies have been identified in the reviewed studies: simulation methods (seven articles) and learning programmes (eight articles). There are also two studies that focus on comparing different teaching methodologies.

#### 3.4.1. Clinical Simulation

The simulation methods focused on in the studies were virtual simulation (based on mobile applications), simulation games, and high-fidelity clinical simulation. Of the total number of nursing students in the studies referring to clinical simulations, 43.85% were in their second year, while 57.1% were senior-year students. The most used method in the clinical simulation group was virtual simulation, and 57.14% of studies included only one-day teaching interventions.

Virtual simulations were used to increase knowledge about medication administration and nasotracheal suctioning in different scenarios [21], to evaluate the effect of interactive nursing skills, knowledge, and self-efficacy [11], and to detect patient deterioration in two different cases [22]. Simulation game methodology was used to improve nursing students’ cognitive and attention skills, strengthen judgment, time management, and decision-making [14].

Clinical simulation was used to develop nursing students’ clinical reasoning in evaluating wounds and their treatments [12], to evaluate and compare the perception of stressors, with the goal of determining whether simulations promote students’ self-evaluation and critical-thinking skills [23], and also to evaluate the impact of multiple simulations on students’ self-reported clinical decision-making skills and self-confidence [24].

#### 3.4.2. Learning Programs

Different types of learning programmes have been identified in this systematic review: team-based learning, reflective training programs, person-centred educational programmes, ethical reasoning programmes, case-based learning, mapping, training problem-solving skills, and self-instructional guides. Of the total number of nursing students in the studies referring to learning programs, 57.1% were junior-year students, while 43.85% were in their senior year.

Team-based learning is a learner-centred educational strategy that promotes active learning to improve students’ problem-solving, knowledge, and practise performance. It can be implemented in small or large groups divided into teams with an instructor and reading material based on case scenarios [25]. Reflective training is based on a new mentoring practise to explore, think about, and solve problems actively during an internship. During the reflective training program, the mentors lead students to uncover clinical nursing problems through conversations with them and discussing feedback for their professional portfolios [26]. The person-centred educational programme focuses on how nursing students perceive individualised care, using design thinking to improve their perception. The use of design thinking gave the students opportunities to apply their theoretical knowledge of the person-centred program to plan innovative solutions that may effectively resolve real-life situations [27]. Another educational programme identified is the ethical reasoning program, and the aim of this is to improve nursing students’ handling of ethical decision-making situations [28], engaging the students in complex ethical clinical situations based on real cases.

Case-based learning was used to explore and demonstrate the feasibility of implementing unfolding cases in lectures to develop students’ critical-thinking abilities [29]. The web-based concept mapping of nursing students was also investigated to determine its impact on critical-thinking skills [30]. Training problem-solving skills were used to find out how it affected the rate of self-handicapping among nursing students [31]. And the last article evaluated the effect of the self-instructional guide to improve clinical reasoning skills on diagnostic accuracy in undergraduate nursing students [32].

## 4. Discussion

Although 158 studies were initially identified, only 15 articles were finally included in this review. The excluded articles were mainly from other disciplines other than nursing and used a less rigorous study design than RCT.

The three longest interventions were developed in Asia [26,28,29]. The longest was 300 h in duration, through one year [30]. These interventions were based on learning programs, case-based learning, person-centred care (PCC), and reflective training programs. However, it is important to take into account that Asian nursing curriculum programmes are different from European or United States curriculum because their internship is carried out only during the last academic degree year, while in Europe, following the European directive 2005/36/CE, 2013/55/UE nursing education requirements of 4600 h (2300 h of clinical practice) is carried out along the 3–4 years of the academic degree [33]. On the other hand, the intervention with the biggest sample was 419 nursing students [30], 210 in the experimental group, and 209 in the control group, and the one with the lowest sample was 51, with 24 students in the control group and 27 in the intervention group [32]. Therefore, all the included studies had a good sample size.

This systematic review has detected different methodologies to help nursing students improve their reasoning and decision-making skills. Virtual simulation was the most frequently used teaching method, both as a mobile application and as a serious game. In terms of its effectiveness in a study carried out in Taiwan, the use of a mobile application resulted in significantly higher knowledge scores, better skill performance, and higher satisfaction in students than traditional paper materials [21]. Virtual simulation [11,14,21] has also proven to be an effective tool for enhancing knowledge and confidence in recognising and responding to rapidly deteriorating patients, but studies that combined two educational strategies were more effective [29], like clinical simulation combined with another teaching strategy such as lectures or videos [12].

An interactive learner-centred nursing education mobile application with systematic contents effectively allowed students to experience positive practical nursing skills [11]. However, in a study comparing serious game simulation versus traditional teaching methods, no significant difference was found immediately or in the month following the training [22], but serious games can improve nursing students’ cognitive skills to detect patient deterioration and to make safe decisions about patient care [14]. Although the innovative teaching method was well received by the students, who expressed higher levels of satisfaction and motivation [22]. We can affirm that the development of a mobile application and its application can be effectively used by nursing students at all levels [11]. However, the performance of all these studies was measured on its short-term outcomes, only 40 min [21], 2 h [22], and 1 week [11,14] of intervention, and was performed with a mean sample size of 97 nursing students.

The data obtained in a study developed in Brazil [12] confirm that clinical simulation is effective for the development of nursing students’ clinical reasoning in wound evaluation and treatment and that clinical simulation in conjunction with other educational methods promotes the acquisition of knowledge by facilitating the transition from what the student knows to rational action. Moreover, the high-fidelity simulation strategy increases the perception of stressors related to a lack of competence and interpersonal relationships with patients, multidisciplinary teams, and colleagues compared with the conventional practice class in the skill laboratory. This increase was related to the students’ capacity for self-evaluation and critical reflection, concerning their learning responsibility and the need to acquire the required skills for patient care [23]. However, in the case of the effect of multiple simulations on students, there are no differences found between the double-versus single-scenario simulations [24]. The intervention time in these three studies was 30 min [23], 3.5 h [12], and 4 days [24]; then the time used to implement the intervention can determine the results obtained.

The different learning methods have an impact on various learning outcomes and students’ variables. Team-based learning [25], reflective training [26], the person-centred education programme [27], web-based concept mapping [30], and teaching cognitive-behavioural approaches [31] have proven to be effective in enhancing problem-solving abilities, knowledge, and reasoning processes and consequently improving the quality of nursing practical education. Team-based learning increased problem-solving ability scores significantly, while those in the control group decreased [25]. Reflective training, developed in China based on the new mentoring approach, was effective in encouraging nursing students to explore, think about, and solve problems actively during an internship, consequently improving their disposition for critical thinking [26]. A person-centred education programme using design thinking can effectively improve how nursing students perceive individualised care. Using design thinking allowed the students to apply their theoretical knowledge of the programme to plan innovative solutions that may effectively resolve real health problems [27]. These programmes were developed in 5 or 6 days [27,31], 1 week or 3 weeks [25,30], and 1 year [26].

The education programme focused on improving ethical decision-making had statistically significant improvements in nursing students’ self-efficacy in communication confidence, complex ethical decision-making skills, and decreased communication difficulty [28]. Case-based learning was more effective with lectures than without them in developing students’ critical thinking abilities [29]. This study was one of the longest developed with 300 h during one school year. This long-term learning intervention could have a positive impact on this study sample. Therefore, the time of the learning intervention could be a limitation in the studied RCTs. The one-time self-instruction guide was ineffective in impacting students’ diagnostic accuracy in solving case studies [32], and it is possible that only one day of intervention is not enough.

Studies have shown that problem- and team-based learning [25,31] are more beneficial than traditional teaching [29], as they enhance nursing skills and improve problem-solving abilities, clinical performance, communication competencies, critical thinking, and self-leadership.

Researchers generally agree that clinical reasoning is an important ability and one of the most important competencies for good nursing practise to ensure optimal patient outcomes [29] and to recognise and address patient deterioration effectively. However, effective communication is crucial in clinical reasoning. It is required to establish a rapport with patients, conduct health evaluations, make collaborative decisions, and discuss clinical cases with colleagues and supervisors. Developing clinical reasoning skills during training is essential to improving nursing professionals’ practice. To enhance clinical reasoning abilities, nursing schools should integrate simulations at every level of education to ultimately improve patient care. Improving nursing students’ preparation will impact the quality of patient care. In addition, new innovative teaching methodologies based on the use of technology could be a motivational driver in nursing clinical reasoning [22].

## 5. Limitations

This systematic review did not perform a search on CINAHL. Although most of the journals included in this database are included in MEDLINE, this should be addressed in the future because of the relevance of the database to nursing research. The results of the included studies could have also been influenced by the different times of the interventions and the different contexts. In addition, the reviewers have identified other studies published in languages other than those required by the inclusion criteria. It seems that many articles are published by Asian researchers, but some of them are not in English, so they cannot be analysed.

## 6. Conclusions

As society progresses, the new generation of nursing students poses a challenge; new technologies are ingrained in their daily lives with access to increasingly advanced technologies like artificial intelligence, and we must adapt training to capture their interest and increase their learning skills. The utilisation of mobile apps, digital simulations, and learning games has a positive impact on the clinical reasoning abilities of nursing students and their motivation. Incorporating new technologies into problem-solving-based learning and decision-making can also enhance nursing students’ reasoning skills. As a result, it is crucial to incorporate these tools into the learning process to maintain students’ interest, motivation, and satisfaction in education. Clinical simulation is particularly important in the training of students in terms of clinical performance. Still, it is necessary to add another teaching method to increase the efficacy of clinical simulations. Therefore, nursing schools should evaluate their current teaching methods and consider integrating or modifying new technologies and methodologies that can help enhance students’ learning, improve their clinical reasoning and cognitive skills, and potentially improve nursing students’ ability to affect patient care positively. By doing so, students will be better equipped to provide high-quality patient care in the future.

## Figures and Tables

**Figure 1 healthcare-12-00090-f001:**
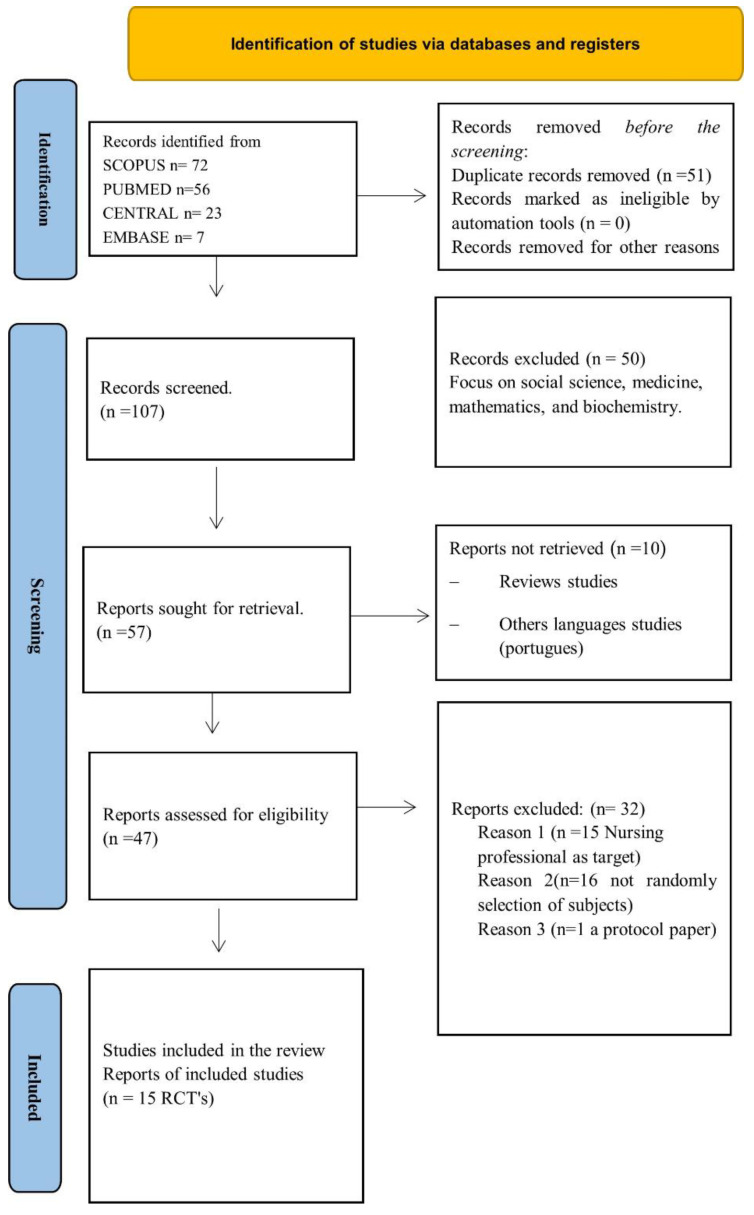
Flowchart of screening of clinical reasoning RCTs that underwent review.

## Data Availability

Data are contained within the article.

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
