# Peer review of "Teaching Strategies for Developing Clinical Reasoning Skills in Nursing Students: A Systematic Review of Randomised Controlled Trials"

_healthcare, 2023, doi:10.3390/healthcare12010090_

Round 1

Reviewer 1 Report (Previous Reviewer 1)

Comments and Suggestions for Authors

Dear authors

The presentation of the manuscript much improved. It is necessary that the results of the included studies be mentioned in detail

Author Response

For research article

Response to Reviewer 1 Comments

1. Summary

2. Questions for General Evaluation

Reviewer’s Evaluation

Response and Revisions

Does the introduction provide sufficient background and include all relevant references?

Yes/Can be improved/Must be improved/Not applicable

Are all the cited references relevant to the research?

Yes/Can be improved/Must be improved/Not applicable

Is the research design appropriate?

Yes/Can be improved/Must be improved/Not applicable

Are the methods adequately described?

Yes/Can be improved/Must be improved/Not applicable

Are the results clearly presented?

Yes/Can be improved/Must be improved/Not applicable

We agree with this comment. Therefore, we have changed the results mentioned more in detail

Are the conclusions supported by the results?

Yes/Can be improved/Must be improved/Not applicable

3. Point-by-point response to Comments and Suggestions for Authors

Comments 1: The presentation of the manuscript much improved. It is necessary that the results of the studies included be mentioned in detail.

Response 1: Thank you for pointing this out. We agree with this comment. Therefore, we have changed the results mentioned more in detail, changes are in red:

3.1 Search results

The initial search identified 158 articles using the above-mentioned strategy (SCOPUS® n=72, PUBMED® n=56, CENTRAL® n=23 and EMBASE® n= 7), and the results are presented in Figure 1. After retrieving the articles and excluding 111, 57 were selected for a full reading. Finally, fifteen papers were chosen following the established selection criteria (Fig.1). The reasons for excluding studies from the systematic review were nurses professionals as targets, other design types of study different that RCT´s, focusing on other health professionals such medical students, review studies, and being published the full text in other languages different that Spanish or English.

                                           Fig.1: Flowchart of screening of clinical reasoning RCTs that underwent review.

 3.2 Risk of Bias CASP results

All studies included in the review were screened with the CASP tool. Each study was scored out of a maximum of 39 points, showing a high quality of the randomised control trial methodology. The studies included had an average score of 33,1, ranging from 30 to 36 points. Besides of this quantitative rate of the items based on CASP, there were 13 studies that missed on item in relation to assessing/analysing outcome/s were 'blinded' or not, and 11 studies that missed the item whether the benefits of the experimental intervention outweigh the harms and costs.

3.3 Data extraction

Once the articles had undergone a full reading and the inclusion criteria were applied, data extraction was performed with a data extraction table (Appendix A). Their contents were summarised into six different cells: 1) CASP total points result, 2) purpose of the study, 3) teaching strategy, 4) time of intervention, 5) sample size, and 6) author and year of publication. After the review by the article's readers, fifteen RCTs were selected. Of the fifteen, the continent with the highest number of studies was Asia, with 53.33 % of the studies (n=8) (Korea n=4, Taiwan n=2 and China n=2), followed by Europe with 26.66 % (n=4) (Turkey n=2, Paris n=1 and Norway n=1), and lastly South America with 20% (n=3), all of them from Brazil.

3.4 Teaching strategies

Different teaching strategies have been identified in the reviewed studies: simulation methods (seven articles) and learning programs (eight articles). There are also two studies that focus on comparing different teaching methodologies.

3.4.1 Clinical simulation

The simulation methods focused on in the studies were virtual simulation (based on mobile applications), simulation games and high-fidelity clinical simulation. Of the total number of nursing students in the studies referring to clinical simulations, 43.85% were in their second year, while 57.1% were senior-year students. The most used method in the clinical simulation group was the virtual simulation and 57.14% of studies included only one day teaching intervention.

Virtual simulations were used to increase knowledge about medication administration and nasotracheal suctioning in different scenarios [21], to evaluate the effect of interactive nursing skills, knowledge and self-efficacy [22], and to detect patient deterioration in two different cases [23]. Simulation game methodology was used to improve nursing students' cognitive and attention skills, strengthen judgment, time manage and decisions making [24].

Clinical simulation was used to develop nursing students’ clinical reasoning in evaluating wounds and their treatments [25], to evaluate and compare the perception of stressors, with the goal of determining whether simulations promote students' self-evaluation and critical-thinking skills [26], and also to evaluate the impact of multiple simulations on students' self-reported clinical decision-making skills and self-confidence [27].

3.4.2 Learning programs

Different types of learning programs have been identified in this systematic review: team-based learning, reflective training programs, person-centred educational program, ethical reasoning program, cased-based learning, mapping, training problem-solving skills and self-instructional guides. Of the total number of nursing students in the studies referring to learning programs, 57.1% were junior-year students while 43.85% were in their senior year.

Team-based learning is a learner-centred educational strategy that promotes active learning to improve students' problem-solving, knowledge and practice performance. It can be implemented in small or big groups divided into teams with an instructor and reading material based on case scenarios [28]. Reflective training is based on a new mentoring practice to explore, think about, and solve problems actively during an internship. During the reflective training program, the mentors lead students to uncover clinical nursing problems through conversations with students and discussing the feedback for their professional portfolios [29]. The person-centred educational program focuses on how nursing students perceive individualised care using design thinking to improve their perception. The use of design‐thinking gave the students opportunities to apply their theoretical knowledge of the person-centred program, to plan innovative solutions that may effectively resolve real situations [30]. Another educational program identified is the ethical reasoning program, and the aim of this is to improve nursing students’ handling of ethical decision-making situations [31],engaging the students in complex ethical clinical situations based on real cases.

The cased-based learning was used to explore and demonstrate the feasibility of implementing unfolding cases in lectures, to develop students’ critical-thinking ability [32]. The web-based concept mapping on nursing students was also investigated, to know the impact on critical-thinking skills [33]. Training problem-solving skills was used to find out how affected the rate of self-handicapping among nursing students [34]. And the last article evaluated was the effect of the self-instructional guide to improve clinical reasoning skills on diagnostic accuracy in undergraduate nursing students [35].

Reviewer 2 Report (Previous Reviewer 2)

Comments and Suggestions for Authors

Noticing that this is a resubmission if a previously rejected article, this research has addressed part of my concerns in the previous review. It appears that the topic of this work is suitable for this special issue.

The introduction section is improved and also the whole article is more completed. The writing style is in general improved as well.

Still, some improvement should be made before accepting this work. Some comments are as follows:

-The concerns regarding the RQ from the last review is not resolved. The research questions should be improved so as to reflect the broader aim of this research with these questions. Meanwhile, some RQ is not addressed throughout the work (E.g. which outcomes were studied?).

- New concerns is developed regarding the figure 1. In the last filter “reports excluded (n=30)” is confusing.

- Given that the authors has incorporated ideas from the other reviewer to include new database and extended the timeframe of review. Still, the same 17 articles was left behind after the whole process, thus the quality of this systematic review is not really improved in terms of the width.

-It is reported that there are studies, both effective ad ineffective reported for two groups. Not sure how did the authors get to the conclusion. In particular, the authors advocate a virtual simulation “Chang er al., 2021”, which only last for 40 mins which perhaps does not really fit into the conclusion. Also, I think the outcome could have been study more in depth. Studies could have been proved effective in some situation, if the authors would like to state one teaching method is as good as the traditional one (line 243), but ineffective if the control group is with no teaching.

Comments on the Quality of English Language

Nil.

Author Response

Response to Reviewer 2 Comments 

1. Summary 

2. Questions for General Evaluation 

Reviewer’s Evaluation 

Response and Revisions 

Does the introduction provide sufficient background and include all relevant references? 

Yes/Can be improved/Must be improved/Not applicable 

Are all the cited references relevant to the research? 

Yes/Can be improved/Must be improved/Not applicable 

Is the research design appropriate? 

Yes/Can be improved/Must be improved/Not applicable 

Are the methods adequately described? 

Yes/Can be improved/Must be improved/Not applicable 

Are the results clearly presented? 

Yes/Can be improved/Must be improved/Not applicable 

Are the conclusions supported by the results? 

Yes/Can be improved/Must be improved/Not applicable 

3. Point-by-point response to Comments and Suggestions for Authors 

Comments 1: The concerns regarding the RQ from the last review is not resolved. The research questions should be improved so as to reflect the broader aim of this research with these questions. Meanwhile, some RQ is not addressed throughout the work (E.g. which outcomes were studied?). 

Response 1: Thank you for pointing this out. We agree with this comment. Therefore, we have changed the RQ from which outcomes were studied? to Which are the purposes of the identified RCT´s? and What were the outcomes of the teaching strategies used in the RCT´s?   Line 73 

Comments 2: - New concerns is developed regarding the figure 1. In the last filter “reports excluded (n=30)” is confusing. 

Response 2: Agree. We have, accordingly, revised and specified more in detail, reason 1: n: nursing professional as target, reason 2 n: not randomly selection of subject and reason 3: protocol paper. See it in Figure 1. 

Comments 3: Given that the authors has incorporated ideas from the other reviewer to include new database and extended the timeframe of review. Still, the same 17 articles was left behind after the whole process, thus the quality of this systematic review is not really improved in terms of the width. 

Response 3: Thank you for pointing this out. We agree with this comment. Therefore, we have made the following changes. The first journal reviewers advised us to focus only on RCTs, so we eliminated quasi-experimental studies. Besides we performed a second review of the literature in the SCOPUS database introducing seven new articles, so the sample size was 17. Moreover, in this last review, we decided to focus only on nursing students, then 2 articles were removed and therefore we had a final sample size of 15 final articles.  

Comments 4: It is reported that there are studies, both effective and ineffective, reported for two groups. Not sure how did the authors get to the conclusion. In particular, the authors advocate a virtual simulation “Chang er al., 2021”, which only lasts for 40 mins which perhaps does not really fit into the conclusion. Also, I think the outcome could have been studied more in depth. Studies could have been proved effective in some situations, if the authors would like to state one teaching method is as good as the traditional one (line 243), but ineffective if the control group is with no teaching. 

Response 4: We also agree with your comments. We have, accordingly, made the following changes. We specify in the discussion section more in-depth which skills improved in nursing students and in which context, line 176 to line 246.  

We also added the outcomes that have proven effective, line 214-221: 

“The different learning methods impact on various learning outcomes and students' variables. Team-based learning [28], reflective training [29] the person-centred education program [30], web-based concept mapping [33] and teaching cognitive-behavioural approaches [34] have proven to be effective in enhancing problem-solving abilities, knowledge and reasoning process, and consequently improve the quality of nursing practical education to. Team-based learning increased problem-solving ability scores significantly while in the control group decreased [28]. Reflective training, developed in China, based on the new mentoring approach, was effective on nursing students to explore, think about, and solve problems actively during an internship and consequently improve their disposition for critical thinking [29]. The person-centred education program using design thinking can effectively improve how nursing students perceive individualised care. Using design thinking allowed the students to apply their theoretical knowledge of the program to plan innovative solutions that may effectively resolve real health problems [30]. These programs were developed in 5 or 6 days [30, 34], 1 week or 3 weeks [28, 33] and 1 year [29]” 

Besides in reference to the effectiveness of the studies we added information about the extend time of the intervention and the sample, line 222-231: 

“The education program focused on improving ethical decision-making had statistically significant improvements in nursing students' self-efficacy in communication confidence, complex ethical decision-making skills and decreased communication difficulty [31]. The case-based learning was more effective with lectures than without them, in developing students' critical thinking abilities [32]. This study was one of the longest developed with 300 hours during one school year. This long learning intervention could have a positive impact in the study sample. Therefore, the time of the learning intervention could be a limitation in the studied RCTs. The one-time self-instruction guide was ineffective in impacting students' diagnostic accuracy in solving case studies [35], and it is possible that only one day of intervention is not enough” 

4. Response to Comments on the Quality of English Language 

The English has been revised use the author services of MPI to improve the quality of the English. 

Round 2

Reviewer 2 Report (Previous Reviewer 2)

Comments and Suggestions for Authors

Thank you for taking my comments into consideration. The discussion is in-depth and convincing. I just wonder if this version provided is the latest since some response to comments in not reflected in this version.

Response 1: the change in RQ is not included in this v2.

Response 2: The last exclusion (n=15) is not shown in anywhere in the v2.

Author Response

v

Response to Reviewer 2 Comments (Round 2)

1. Summary

Thank you very much for taking the time to review this manuscript and help us to improved it. Please find the detailed responses below and the corresponding revisions in the re-submitted files.

2. Point-by-point response to Comments and Suggestions for Authors

Thank you for taking my comments into consideration. The discussion is in-depth and convincing. I just wonder if this version provided is the latest since some response to comments in not reflected in this version.

Comments 1: The change in RQ is not included in this v2.

Response 1: Thank you for pointing this out. We agree with this comment. We have had problems with the web page and maybe we didn´t upload the latest version. Therefore, we check it, and we can confirm that the RQ changes are in the line 87 to 90.

Which randomised controlled trials have been conducted in the last eleven years regarding nursing students' clinical reasoning? Which are the purposes of the identified RCT´s? Which teaching methodologies or strategies were used on the RCT´s studies? What were the outcomes of the teaching strategies used in the RCT´s?  

Comments 2: The last exclusion (n=15) is not shown in anywhere in the v2.

Response 2: We have, accordingly revised, and modified. Line 156 to 161

To be consisting with the methodology the independents reviewers analysed all the selected articles one more time after the additional search, and they agreed to eliminate two of them, because the study sample included nursing students but also professional nurses. Therefore, to have a clear outcome, focused on nursing students, two articles were removed and the very final sample size was of fifteen articles, following the established selection criteria (Fig.1)

This manuscript is a resubmission of an earlier submission. The following is a list of the peer review reports and author responses from that submission.

Round 1

Reviewer 1 Report

Comments and Suggestions for Authors

Dear authors

I would like to thank you for giving me the opportunity to review the manuscript entitled " Clinical reasoning in nursing students-educational strategies”. As a systematic review, this study does not have the necessary standards, it has very poor writing, and the synthesis of the results is not done well. I won't go into too much detail and will make some points:

Title

1. Systematic review should be identified in the title.

Abstract

2. Abstract is not informative. Please follow the PRISMA 2020 for Abstracts.

Methodology

3. According to the number of retrieved studies, it seems that the process of selecting keywords and searching has not been done well. Why isn't Scopus used for searching?

4. A search query must be provided for each database separately.

5. I suggest to use appropriate sub-headings for this section: Protocol and registration, Search process and eligibility criteria, Study selection, Risk of bias / quality assessment, Data collection process and synthesis of results

6. Why is the risk of bias of the studies not evaluated?

7. The figure is very messy.

Results

8. The synthesis of studies is not well done and is not informative.

Conclusion

9. What about conclusion?

Reviewer 2 Report

Comments and Suggestions for Authors

This research provides a critical review of the past studies regarding the education strategies to nurture clinical reasoning in nursing students. It appears that the topic of this work is suitable for this special issue. Still, significant improvement should be made before accepting this work. Some comments are as follows:

-English standard of this work should be improved. The title is not clear as well.

-The introduction is not well constructed. The context, necessity of this research is not explained clearly. This part regarding detailed descriptions of different teaching strategies seems to be “artificial” as it correctly maps with the results.

-The research questions should be improved so as to reflect the broader aim of this research with these questions. Meanwhile, some RQ is not addressed throughout the work (E.g. which outcomes were studied?).

-Not sure why it is empty in section 3?

-The description of line 131 (25 articles did not meet the selection criteria) seems not to be correct.

-It seems that only the authors only report the teaching strategies in section4 while the effectiveness is not mentioned. Therefore, it is hard to come up to conclusion in Section 5, that problem based learning is more effective. Meanwhile, the time of intervention should be took into consideration when such judgement is made.

- the discussion part does not really integrate all the results. In most cases, it just repeat individual results of one of the article identified.

Comments on the Quality of English Language

The English standard could be further improved. It is suggested to unify the use of tense used in each section. Also, the title could be further improved.

Reviewer 3 Report

Comments and Suggestions for Authors

dear author.

Thank you for reading. 

I wanted to congratulate you on your work. You have done a very good job. However, I would like to make a number of clarifications that I think would improve the quality of your report.

- The inclusion criteria are described in a veiled manner in the text. Could you make them explicit, as well as the exclusion criteria?

- You adequately describe the process of double-blind reading of the abstracts of the initially selected research papers and the establishment of a scoring process with CASPE for the finally selected papers. In this regard:

-- Could you indicate the criteria for inclusion/exclusion of a paper in case of conflicting opinions? Was there 100% agreement between the researchers on the papers included/excluded? 

-- The results of the CASPE surveys should be included in the results section, not in the methodology section, as it does not condition the selection process but illustrates the quality of the selected papers. 

-- Could you indicate the degree of consensus reached by the researchers when rating the papers evaluated with CASPE? Are the CASPE scores given as a result of the papers analysed the average of the two raters? 

--Indicate the identity of the researchers who carried out this process. Were they the authors of the work? Did any external researcher participate?

- The boxes in Figure 1 should be improved. They are partially legible.

- The Discussion section offers a reflection of the results found in your research, and denotes an apparent comparison of the results of your work with other similar work or aspects to be taken into account. Please modify this section accordingly. 

- 33% aprox of the papers referenced are more than 5 years old. You could replace them with more up-to-date references. Please justify it conveniently. 

Kinds Regards

Reviewer 4 Report

Comments and Suggestions for Authors

Thank you for the opportunity to review your article, I agree that clinical reasoning is a vital skill for health professionals to learn and that with some variations in your article you will be able to articulate this as well.  I do find that, which an important topic, you have not clearly demonstrated the importance in your article introduction.  This may be a matter of language and article structure though and I believe you can make this clear with revisions.  I do also have a number of other comments which I hope you will find helpful in a subsequent draft submission.

Thank you again and I hope to be able to review a later submission with corrections.

Comments on the Quality of English Language

There are some sections where English language expression was in need of correction, however there are others where language was easy to follow and read.  I would encourage you to seek some editorial support on English language expression if it is not your first language.